# Bioactive Peptides from Edible Mushrooms—The Preparation, Mechanisms, Structure—Activity Relationships and Prospects

**DOI:** 10.3390/foods12152935

**Published:** 2023-08-02

**Authors:** Haiyan Li, Ji’an Gao, Fen Zhao, Xinqi Liu, Biao Ma

**Affiliations:** 1Key Laboratory of Geriatric Nutrition and Health, Beijing Advanced Innovation Center for Food Nutrition and Human Health, National Soybean Processing Industry Technology Innovation Center, Beijing Technology and Business University, Beijing 100048, China; l15069433928@163.com (H.L.); gja200828@163.com (J.G.); liuxinqi@btbu.edu.cn (X.L.); 2Beijing Science Sun Pharmaceutical Co., Ltd., Beijing 100176, China; mabiao@ssyy.com.cn

**Keywords:** edible mushrooms, preparation, bioactivities, mechanisms, structure–activity relationships, functional food

## Abstract

Mushroom bioactive peptides (MBPs) are bioactive peptides extracted directly or indirectly from edible mushrooms. MBPs are known to have antioxidant, anti-aging, antibacterial, anti-inflammatory and anti-hypertensive properties, and facilitate memory and cognitive improvement, antitumour and anti-diabetes activities, and cholesterol reduction. MBPs exert antioxidant and anti-inflammatory effects by regulating the MAPK, Keap1-Nrf2-ARE, NF-κB and TNF pathways. In addition, MBPs exert antibacterial, anti-tumour and anti-inflammatory effects by stimulating the proliferation of macrophages. The bioactivities of MBPs are closely related to their molecular weights, charge, amino acid compositions and amino acid sequences. Compared with animal-derived peptides, MBPs are ideal raw materials for healthy and functional products with the advantages of their abundance of resources, safety, low price, and easy-to-achieve large-scale production of valuable nutrients for health maintenance and disease prevention. In this review, the preparation, bioactivities, mechanisms and structure–activity relationships of MBPs were described. The main challenges and prospects of their application in functional products were also discussed. This review aimed to provide a comprehensive perspective of MBPs.

## 1. Introduction

The hydrolysis of proteins by proteases produces short-sequence amino acids with special physiological functions called bioactive peptides [1]. Bioactive peptides have mostly been isolated from a variety of animals and plants. Bioactive peptides are often obtained from animal sources such as milk and dairy products [2], eggs [3], meat [4] and fish-derived products [5]. Bioactive peptides from plants are mainly obtained from legumes [6], grains [7], nuts [8], fruits and vegetables [9]. Until now, a significant amount of research about bioactive peptides isolated from animals and plants has been intensively reported, while relatively few studies have been conducted on edible mushrooms bioactive peptides (MBPs). Currently, there are at least 12,000 species of mushrooms, of which 2000 are reported to be edible, and about 200 are collected as food or pharmaceutical ingredients [10]. Therefore, edible mushrooms are rich and vast in relatively untapped resources. Different bioactive compounds isolated from different species of mushrooms can be used to develop different functional foods. Edible mushrooms are excellent functional foods and nutritional supplements due to their high quantities of bioactive metabolites, including proteins, polysaccharides, enzymes (e.g., superoxide dismutase), dietary fibre and many other biomolecules [11]. For example, *Trametes versicolor* are commonly marketed as dietary supplements in the form of tablets, capsules and powders [12]. Recently, the natural flavour compounds in *Trametes versicolor* have been valuable ingredients in the production of non-alcoholic or low-alcoholic beers [13,14]. Based on US Department of Agriculture (https://fdc.nal.usda.gov/ (accessed on 8 July 2020)) statistics, it can be seen that edible mushrooms contain higher protein than most vegetables, and many proteins have bioactivities, while more or even new MBPs remain to be discovered [15,16]. MBPs can be absorbed by the intestine completely, producing direct partial benefits in the digestive tract without increasing the functional burden on the gastrointestinal tract [1]. MBPs can exert beneficial physiological effects by entering the circulatory system in their intact state. MBPs can provide high organisational affinity [17], low toxicity and high stability [18], with a range of nutritional, functional and biological properties. Endogenous MBPs may exert antioxidant and anti-inflammatory activities by regulating antioxidant pathways and activating immune cells. MBPs may also act as analgesics or opioids in improving memory and cognitive deterioration in nerve cells [19]. Exogenous MBPs may be used as functional foods and medicines to ameliorate inflammation, hyperglycemia, hypertension, high cholesterol and other degenerative diseases [20]. Recent research found favourable health effects from *Agaricus bisporus* bioactive peptides as new generations of prebiotic and probiotic microorganisms in synbiotic preparations [21]. For consumers who prefer not to consume animal-derived products, MBPs can be sources of ideal alternative supplements for replacing animal-derived food [22]. Replacing some of the meat or fish with healthy natural foods is a trend for the future market. Edible mushrooms have a similar taste to meat products and are rich in many nutrients. For this reason, many entrepreneurs have incorporated edible mushrooms into muscle foods (e.g., meat and fish) to reduce the proportion of meat [23].

Thus far, many studies on peptides have focused on those isolated from animals and plants, while relatively few reviews have been conducted on MBPs. Thus, this review will discuss the preparation, bioactivities and mechanisms and structure–activity relationships of MBPs, as well as their applications in functional foods.

## 2. Preparation of MBPs

Edible mushrooms are important sources for naturally bioactive proteins and peptides, which provide an excellent material basis for the discovery of MBPs. The protein content of edible mushrooms by-products is about 25%, and bioactive peptides extracted from edible mushrooms by-products can be recycled for the production of functional foods, which can effectively reduce the waste of resources [24]. The extraction methods are diverse because of the different growing environments and different species of edible mushrooms. As a result, the structures and bioactivities of MBPs also vary [9]. Basically, there are two popular approaches to prepare MBPs; one is to extract endogenous MBPs directly from the mushrooms. Many endogenous MBPs have been extracted directly from fresh fruiting bodies, dried powders or fermented powders, such as peptides with antimicrobial and ACE inhibitory properties, respectively [25]. The other approach uses proteolytic reactions of exogenous enzymes to release peptide fragments from mushroom proteins isolated from edible mushrooms or their mycelia indirectly (e.g., hydrolysis by bromelain) to liberate potent peptides from intact proteins [9]. Indeed, MBPs with a variety of biological activities, such as antioxidative, antibacterial, anti-inflammatory, anti-aging, antitumour and anti-diabetic activities, were obtained using enzymatic hydrolysis, for example, peptides with ACE inhibition hydrolysis from *Lentinus edodes* with Alcalase [26]. MBPs contain ergothioneine, laccase, ribonuclease and other enzymes [27]. Low molecular weight MBPs obtained with the assistance of ultra-high-pressure processing were found to activate ethanol dehydrogenase and aldehyde dehydrogenase in vitro. They could effectively catalyse the gradual conversion of alcohol into acetaldehyde and acetic acid with lower toxicity, and prevent alcoholic liver injury [15].

Chemical hydrolysis is also one of the commonly used methods to prepare MBPs. Chemical hydrolysis involves breaking the peptide chains of a protein using an acid or base solution, producing peptides and free amino acids. Chemical hydrolysis in the food industry has a variety of limitations, such as difficulties controlling the process, loss of nutrients and pollution of the environment [28]. Microbial fermentation is also one of the commonly used methods to obtain MBPs in recent years. Microorganisms have protein hydrolysis systems that can produce proteases or induce fermentation to obtain MBPs. The type of MBPs depends on the fermentation time, the strain and the type of edible mushrooms protein [29]. Microbial fermentation can avoid impurities in peptides produced by enzymatic digestion, shorten preparation time and reduce production costs.

Ultrafiltration, hydrophilic interaction chromatography (HILIC), fast protein liquid chromatography (FPLC), ion exchange chromatography (IEC), size exclusion chromatography (SEC), gel filtration chromatography (GFC) or high-performance liquid chromatography (HPLC) [25] are methods commonly used for purification after extraction and preliminary bioactivity screening. For example, antihypertensive peptides were separated and characterised from the *Agaricus bisporus* using SEC, reversed-phase-high-performance liquid chromatography (RP-HPLC) and liquid chromatography–mass spectrometry (LC–MS/MS) [25].

MBPs alleviate symptoms associated with targeted damage or degenerative diseases. In addition, MBPs are known to exert antioxidant effects by inactivating reactive oxygen species and scavenging free radicals to improve degenerative diseases such as inflammation, hypertension, hyperglycemia, cardiovascular disease, high cholesterol, memory and cognitive diseases with antioxidant, antibacterial, anti-aging and antitumour effects (Figure 1).

## 3. Bioactivities of MBPs

### 3.1. Antioxidant Activity

Edible mushrooms are rich in a variety of peptides with antioxidant activity [30]. Due to the different raw ingredients and preparation processes, MBPs may have different targeting functions. The antioxidant mechanisms of MBPs are mainly categorised into those involving regulation of ROS production, regulation of antioxidant enzyme activities and regulation of antioxidant pathways. Free radicals may be scavenged by MBPs through providing protons, electrons and chelating metal ions to regulate the production of ROS [31]. Low molecular weight peptides isolated from *Agaricus bisporus* (ABP) and *Pleurotus eryngii* mycelium (PEMP) were abundant in negatively charged amino acids, which could neutralise free radicals and regulate the production of ROS [32]. G. lucidum peptide (GLP) [33] exerted antioxidant effects in the soybean oil system by blocking soybean lipoxygenase activity in a dose-dependent manner, with an IC_50_ value of 27.1 µg/mL. Compared to butylated hydroxytoluene, GLP had the better antioxidant activity through scavenging hydroxy radicals and quenching superoxide radical anions in biological systems.

Organisms can regulate the production of ROS in time, through endogenous enzymes and non-enzymatic defensive systems [34]. The oral administration of GLP exerted significant hepatoprotective effects through antioxidant activity in mice with liver injuries [35]. GLP increased the levels of glutathione (GSH) and superoxide dismutase (SOD), decreasing the levels of malondialdehyde (MDA) in the liver. Furthermore, it decreased the activities of alanine transaminase (ALT) and aspartate transaminase (AST) in the serum to resist hepatic fibrosis and alcoholic liver injury. Research showed that GLP could effectively inhibit the hazards of peroxide produced in mitochondria by regulating the activity of antioxidant enzymes, showing excellent antioxidant activity.

MBPs achieve antioxidant activity by modulating antioxidant pathways such as the Kelch-like ECH-associated protein 1-nuclear factor erythroid-2-related factor 2-antioxidant responsive elements (Keap1-Nrf2-ARE), nuclear factor-κ light chain enhancer of activated B cells (NF-κB), mitogen-activated protein kinase (MAPK) and the phosphoinositide 3-kinase/protein kinase B (PI3K/AKT) pathways [36,37,38]. MBPs may regulate the expression of related antioxidant proteins by down-regulating the Keapl gene and up-regulating Nrf2 gene expression (Figure 2) [31]. The GLP [39] promoted Nrf2 and activated the Nrf2-ARE signalling pathway, exhibiting antioxidant effects on cells induced by hydrogen peroxide (H_2_O_2_). Nrf2 dissociated from Keap1 upon stimulation of the PI3K/AKT and MAPK pathways, moving to the nucleus, and associated with the antioxidant component ARE, which regulated the expression of antioxidant enzymes such as HO-1, catalase (CAT), etc. [40].

#### Anti-Aging Activity

Aging occurs in cells, organs and the whole organism, which leads to a reduction in biological function of the organism’s ability to eliminate oxidative stress. The accumulation of excess free radicals causes MDA levels to increase, the total antioxidant capacity (T-AOC) to reduce, and disrupt cellular structure [41], leading to cellular senescence and death. CMP [42] and GLP [43] had dose-dependent scavenging effects on oxygen free radicals and hydroxyl free radicals. SOD is a critical mitochondrial enzyme antioxidant associated with longevity [44]. MBPs could significantly enhance senescence-associated mitochondrial enzyme antioxidants, as displayed in Table 1. CMP and GLP scavenged hydroxyl radicals better than the specific hydroxyl radical scavenger mannitol [45]. Agaricus blazei peptide (ABp) reduced MDA and ROS contents, and increased CAT and T-AOC activities in a D-galactose-induced senescence model of the NIH/3T3 cell [46].

The main targets of MBPs to slow down the aging process include the metabolic mitochondrial pathway, inactivating reactive oxygen species, scavenging free radicals, alleviating the oxidation of biomarkers in organisms, and reconstructing homeostatic mechanisms in vivo [47]. Nrf2 activity was tightly correlated with degenerative diseases induced by aging, and contributed to the prevention and mitigation of degenerative diseases. In the aging model established by D-galactose in mice, feeding ABp was found to lead to the down-regulation of Keap1 protein expression, thus up-regulating Nrf2. In the Keap1–Nrf2 pathway, the expressions of HO-1 and related factors such as ApoE, Hsph1 and Trim32 were up-regulated, effectively scavenging free radicals and showing excellent anti-aging activity [48]. Progressive changes were found in the epigenetic information in both dividing and non-dividing cells in modifications involving chromatin transformations, histone and DNA methylation patterns in research related to aging. The major features of aging are constituted in DNA and histones methylation, accompanied by other epigenetic alterations [49]. In previous studies, it was found that the epigenetic alterations of aging could be modulated by the consumption of peptides that directly maintain telomere length [44]. Current research about whether MBPs could regulate telomere length in different organisms has not been conducted. This would be a new area of research on the anti-aging activities of MBPs, and more relevant research is needed in the future.

In conclusion, MBPs can act on multiple targets simultaneously to exert their functional antioxidant and age-delaying activities. MBPs have a wide scope for development as one of the components of natural antioxidant and anti-aging functional foods. Currently, MBPs have become a new direction of research on functional foods. However, further research is needed on how MBPs replace synthetic antioxidants.

**Table 1 foods-12-02935-t001:** Anti-aging peptides derived from edible mushrooms.

Edible Mushroom Category	Molecular Weight and Amino Acid Sequence	Mechanisms of Operation and Value	Reference
*Agaricus blazei*	Peptide mixtures	Alleviated D-gal-induced senescence response in NIH/3T3 cells. Decreased MDA and ROS contents Increased SOD, CAT and T-AOC activities	Feng et al. (2020) [46]
*Ganoderma lucidum*	Peptide mixtures	Inhibited expression of NOX4, TGF-β1 and 8-hydroxy-2′-deoxyguanosine (8-OHdG) Decreased MDA and ROS contents Increased SOD activities	Meng et al. (2022) [50]
*Agaricus bisporus*	MW of 1~3 kDa	DPPH radical scavenging activity with IC_50_ of 0.13 mg/mL Reduction in lactate dehydrogenase (LDH) leakage Decreased MDA and ROS contents Increased CAT and GSH activities	Kimatu et al. (2020) [51]
*Tricholoma matsutake*	Amino acid sequence: WALKGYK, WFNNAGP	Alleviated oxidative damage in DSS-induced mice DPPH radicals scavenging activity is 50% at 10 mg/mL Increased SOD contents Decreased MDA contents	Geng et al. (2016) [52,53]
*Tricholoma matsutake*	MW of 1~3 kDa, Amino acid sequence: EHEEHEEHEEPEDDPNSSEESYW	DPPH radical scavenging activity is 70% With EC 50 of 0.468 mg/mL Decreased MDA and ROS contents Increased CAT and T-AOC activities	Feng et al. (2020) [48]

### 3.2. Antimicrobial Activity

Antibiotics can treat infections such as tuberculosis, pneumonia, leprosy and gonorrhoea. However, antibiotic resistance [54,55] has emerged from the overuse and misuse of antibiotics. Natural bioactive peptides are characterised by their high efficacy, stability and low toxicity, making them major alternatives to antibiotics and conventional drugs [56]. MBPs have been widely studied for their antimicrobial activities. A variety of antimicrobial peptides have been isolated and purified from edible mushrooms such as *Polyporus alveolaris*, *Pleurotus eryngii* [57], *Lentinus edodes* and *Agrocybe cylindracea* [58] (Table 2). The possible antimicrobial mechanisms of MBPs could involve either regulating and leading to tissue-specific expression patterns, or making intracellular protein leakage and leading to bacterial death.

The hydrophobic amino acids of MBPs could regulate the NF-κB pathway and the MAPK pathway. PEMP [32] had good antibacterial activity by stimulating macrophage proliferation, increasing phagocytosis activities, TLRs expression, and releases of tumour necrosis factor-α (TNF-α), IL-6, NO and H_2_O_2_. MBPs are rich in structures with α-helices, β-folds, random coils and disulfide bonds [59]. Antimicrobial peptides are enriched with hydrophobic amino acids, α-helixes, β-folds, random coils and disulfide bonds.

Another antibacterial mechanism of MBPs is the disruption of bacterial cell membranes, causing intracellular protein leakage within the bacterial cells to achieve antibacterial effects. Antimicrobial peptides from the mycelia of *Cordyceps militaris* [60] may cause intracellular protein leakage of E. coli (ATCC 25922), which maintained the integrity of the intestinal mucosa and attenuated E. coli infections in mice. Antibacterial peptides isolated from the mycelia (GLM) and fruiting bodies (GLF) of G. lucidum [61] showed a dose-dependent increase at 50–125 μg/mL in protein leakages from *Escherichia coli* and *Staphylococcus aureus*. It is clear that GLF and GLM induces cell death with strong antibacterial activities against both *Escherichia coli* and *Staphylococcus aureus*.

Based on the above, MBPs show good antibacterial activities against bacteria that are resistant to drugs. Thus, MBPs could be important natural alternatives to antibiotics.

**Table 2 foods-12-02935-t002:** Antimicrobial peptides derived from edible mushrooms.

Edible Mushroom Category	Molecular Weight and Amino Acid Sequence	Inhibited Bacterial Species and Inhibition Values	Reference
*Polyporus alveolaris*	MW of 28 kDa	*F. oxysporum*, *Physalospora piricola*, *M. arachidicola*, *Botrytis cinereal*, with inhibitory concentration of 8 mg/mL for all	Wang et al. (2004) [62]
*Pseudoplectania nigrella*	MW of 4398.80 Da	*Streptococcus pneumoniae* in mice, with inhibitory concentration of 10 mg/kg	Mygind et al. (2005) [63]
*Agaricus bisporus*	Peptide mixtures	*Pseudomonas aeruginosa*, reduced to 26.64% at 0.25 mg/mL	Farzaneh et al. (2018) [64]
*Terfezia claveryi*	Peptide mixtures	*Bacillus cereus*, reduced to 27.44% at 0.25 mg/mL	Farzaneh et al. (2018) [64]
*Ganoderma lucidum*	Peptide mixtures	*Salmonella typhi* with minimal inhibitory concentration (MIC) of 52 μg Escherichia coli with MIC of 60 μg	Mishra et al. (2018) [18]
*Pleurotus ostreatus*	MW of 7 kDa N-terminal: VRPYLVAF	*F. oxysporum*, reduced to 20% at dosage of 15.6 μM *M. arachidicola*, reduced to 45% at dosage of 15.6 μM *P. piricola*, reduced to 63% at dosage of 15.6 μM	Chu et al. (2005) [65]
*Agrocybe cylindracea*	MW of reduced 9 kDa N-terminal: ANDPQCLYGNVAAKF	Human immunodeficiency virus type 1 with IC_50_ of 60 μM *F. oxysporum* with IC_50_ of 125 μM *M. arachidicola* with IC_50_ of 60 μM	Ngai et al. (2005) [58]
*Cordyceps militaris*	MW of 10.9 kDa N-terminal: AMAPPYGYRTPDAAQ	*Bipolaris maydis* with IC_50_ of 50 μM *Mycosphaerella arachidicola* with IC_50_ of 10 μM *Candida albicans* with IC_50_ of 0.75 mM *Rhizoctonia solani* with IC_50_ of 80 μM Human immunodeficiency virus type 1 with IC_50_ of 55 μM	Wong et al. (2011) [66]
*Lentinus squarrosulus*	MW of 17 kDa	*Trichophyton mentagrophytes* with inhibition zone diameter of 25.7 mm *T. rubrum* with inhibition zone diameter of 22.8 mm *Aspergillus niger* with inhibition zone diameter of 12.64 mm *Candida tropicalis* with inhibition zone diameter of 20.54 mm *C. albicans* with inhibition zone diameter of 20.62 mm	Poompouang and Suksomtip (2016) [67]
*Russula paludosa*	MW of 4.5 kDa N-terminal: KREHGQHCEF	Human immunodeficiency virus type 1 with IC_50_ of 11 μM	Wang et al. (2007) [68]

### 3.3. Anti-Inflammatory Activity

MBPs may enhance the cytotoxicities of natural killer cells and the phagocytosis of macrophages, promoting multiplication and maturation of immune cells and lymphocytes and inhibiting pro-inflammatory responses, thereby improving the host’s defence against invading pathogens (Figure 3) [69]. Yu [70] prepared a bioactive peptide, KSPLY, with a molecular weight of 608.3834 Da, from *Hericium erinaceus*. KSPLY promoted TNF-α, NO, IL-6 and IL-1β secreted by macrophages, which inhibited lipopolysaccharide (LPS)-induced inflammatory responses at a concentration of 100 μmol/L. The proliferation of splenic lymphocytes was significantly reduced in mice fed with *Pleurotus eryngii* peptide (PEP), and the serum haemolysin level of CTX-induced mice was significantly increased [71]. This confirmed that PEP could significantly improve humoral immune function of immunosuppressed mice.

MBPs may enhance the oxidative defensive system and barrier function by promoting the production of antibodies, cytokines and chemokines to reduce the inflammatory response. After local nasal immunotherapy (LNIT) with *Flammulina velutipes* peptides (FIP-fve), the production of proinflammatory cytokines and chemokines were significantly reduced [72]. It has been shown that FIP-fve induces the potent activator of peripheral lymphocytes through activation of the p38 mitogen-activated protein kinase (p38 MAPK) signalling pathway with anti-inflammatory activity. The *Tricholoma* matsutake-derived peptide WFNNAGP [53] attenuated the inflammatory response by inhibiting the expression of pro-inflammatory cytokines and myeloperoxidase (MPO), as it promoted expressions of tight junction proteins closing ribbon-1, claudin and occluding. WFNNAGP reduced colonic inflammation in mice by down-regulating NF-κB expression to inhibit formation and activation of NLRP 3 and caspase-1.

MBPs have a variety of target cells and sites, such as NK cells, CD4^+^, CD25^+^, T lymphocytes, macrophages [73], monocytes, B lymphocytes [74] and mast cells [75], among others. The results of a study showed that FIP-fve could make changes in Treg-associated immunity, with down-regulation of IL-4^+^/CD4^+^ T-cell expression and up-regulation of IFN-γ^+^/CD4^+^ T-cell expression in mice. FIP-fve effectively decreased inflammatory cell infiltration and epithelial damage [72]. Experiments demonstrated that the anti-inflammatory effect of orally administering FIP-fve on mite-induced airway inflammations in mice. Bioactive polypeptides of *Cordyceps militaris* (CMP) [76] were implicated in the regulation of immune function in mice through transcription factors Ets1 and the Spp1, Rel, and Smad3 genes. CMP regulated TNF and the PI3K-Akt signalling pathway, playing an important role in inflammation by increasing immune organ indexes, the number of leukocytes and the content of hemolysin in the sera of mice. In summary, MBPs have good anti-inflammatory activity, and offer new ideas for the development of functional food supplements as natural ingredients.

### 3.4. Memory and Cognitive Improvement Activity

Medications that are commonly used clinically to prevent and improve learning in learning and memory impairments include free radical scavengers, acetylcholinesterase (AChE) inhibitors and drugs to prevent the formation of amyloid β deposits [77]. However, these types of drugs are associated with adverse side effects, high toxicity, poor memory improvement and no preventive effect on the onset and development of neurological disorders. CMP scavenges oxygen free radicals, decreasing AChE activity in the mouse brain, which may reduce the damage of cholinergic neurological nerves, hence improving learning and memory ability in a scopolamine-induced mice model of learning and memory impairment [78]. The mechanism of CMP is similar to that of free radical scavengers and AChE inhibitors. CMP promotes the expressions of Il-1β and Slc18a2, as well as secretion of neurotransmitters in mice that reduce dyskinesia and depression-like behavior, and contributed to improve learning and memory. Hericium erinaceus [79] fed to mice prevented the loss of spatial short-term and impairments of visual recognition memory induced by the formation of amyloid β deposits. However, there are relatively few studies on improvements in memory and cognitive ability by *Hericium erinaceus* peptides, and more experiments are needed to verify its mechanism in the future.

Based on the above, MBPs may effectively prevent and improve learning and memory disorders, with have the advantages of fewer side effects, less toxicity, more targets of action and lower costs. MBPs may be more effective in preventing memory and cognitive disorders compared with synthetic drugs.

### 3.5. Anti-Hypertensive Activity

Hypertension is currently one of the most common metabolic diseases, and is a predisposing factor for many other conditions such as renal failure and heart disease. The inhibition of angiotensin-converting enzyme (ACE) regulation of the renin–angiotensin system is thought to be the main mechanism of hypotensive activity [80]. Edible mushrooms are good sources of hypotensive peptides, and many antihypertensive peptides have been isolated (Table 3). Kaprasob [81] found that LIYAQGFSK peptide extracted from the King Boletus mushroom had the greatest ACE-binding energy of −9.2 kcal/mol through hydrogen bonds, and exhibited ACE inhibitory activity. The AHEPVK, RIGLF and PSSNK peptides of ABP showed stronger ACE inhibitory activities with lower IC_50_ values after gastrointestinal digestion compared to ACE inhibitory peptides (ACEIPs) from other sources [25]. MBPs inhibit ACE mainly through the inhibition of nucleic acid synthesis or binding, blocking protein synthesis, membrane permeabilization, inhibition of enzyme activities and triggering apoptosis. Gln-Leu-Val-Pro (QLVP) is a polypeptide with its amino acid sequence isolated from the mycelia of *Ganoderma lucidum*. QLVP significantly activated the angiotensin I-mediated phosphorylation of endothelial nitric oxide synthase in human umbilical vein endothelial cells, and partially reduced mRNA and protein expressions of vasoconstrictor factor endothelin-1, which showed value in the application of GLP for treating hypertension in metabolic diseases. Overall, MBPs exhibit good anti-hypertensive activities as potential ingredients for natural anti-hypertensive functional foods and nutraceuticals.

### 3.6. Antitumour Activity

Biological therapy is an extensively alternative approach to limit the growth of cancer cells that uses biologically active compounds to treat cancer. Bioactive peptides in food can inhibit cancer at all stages of the disease, and offer advantages such as their greater affinity, target-specific effects, reducing toxicity and superior tissue penetration, in comparison to the side effects of chemotherapeutic molecules [87]. PEP was found to inhibit the growth of cancer cells in concentrations of 0.05 to 2 mg/mL. It inhibited the growth of gastric cancer (HGC-27), breast cancer (BT-549) and cervical cancer (Hela-229) cells by 61.40%, 59.20%, and 32.80%, respectively, and demonstrated strong antitumour activity. The MBPs exerted anti-tumour activity by stimulating the host′s defensive mechanism to enhance its non-specific immune responses. CMP [42] activated the reticuloendothelial system and macrophages to promote lymphocyte transformation and immunologically active cells like lymphocytes, lymphokines, monocyte macrophage systems and NK cells to attack target cells to have anti-tumour effects. The amino acid sequence of the Ser-Leu-Ser-Leu-Ser-Val-Ala-Arg peptide extracted from morels (*Morchella* spp.) were found to reduced cell proliferation via a mitochondrial-dependent pathway. It was shown that the movement of cytochrome C from the mitochondria to the cytoplasm could be facilitated by the down-regulation of Bcl-2/Bax, thereby promoting the expression of caspase-9 and caspase-3 and suppressing tumourigenesis [88]. MBPs facilitated intermediary metabolism or controlled DNA transcription and translation through the activation of related enzyme systems. Mice fed with CMP [89] were found to significantly inhibit polymorphonuclear cells infiltration and the infrared ray-induced up-regulation of brain production of C3 protein levels, and the production of interleukin-1β and tumour necrosis factor-α. MBPs exhibit antitumour effects, and may be promising natural tumour preventive agents.

### 3.7. Other Activities

In addition to their antioxidant, antibacterial, anti-inflammatory, anti-aging, memory and cognitive improving, anti-hypertension and anti-tumour activities shown above, MBPs also have functional activities, such as reducing the levels of plasma glucose with induced diabetes, and decreasing cholesterol levels.

Diabetes is a metabolic disease that is characterised by hyperglycaemia. Prolonged uncontrolled hyperglycaemia in an organism can generate oxidative stress, promote apoptosis, activate protein kinase isozymes and transcription factors, and other pathophysiological mechanisms that impair the proper functioning of tissues and organs [90]. Currently, glycaemia is mainly controlled by the intramuscular injection of hormones or taking synthetic anti-diabetic agents such as insulin, thiazolidinediones, dipeptidyl peptidase 4 inhibitors, biguanides, sulfonylureas and sodium–glucose cotransporter type 2 inhibitors [91]. These hypoglycaemic agents are highly effective, but with side effects [92]. Compared with other synthetic hypoglycaemic substances, MBPs show fewer side effects, and may regulate glycaemia by inhibiting the activities of α-amylase, α-glucosidase and dipeptidyl peptidase-IV. At 2.0 mg/mL, a morel peptide inhibited α-amylase and α-glucosidase by 34.93% and 30.56%, respectively [93], showing excellent anti-diabetic activity.

Hypercholesterolaemia arises from increased endogenous cholesterol synthesis or an excess supply of dietary cholesterol or cholesterol precursors [94]. Bioactive peptides may inhibit the synthesis and secretion of triglycerides and cholesterol by stimulating bile acid secretion from the gallbladder, modulating hormones and cholesterol receptors, and altering hepatic lipid metabolism [94]. MBPs are rich in inhibitors of HMG-CoA reductase, a pivotal enzyme in endogenous cholesterol biosynthesis, as well as ergocalciferol, eritadenine and β-glucan derivatives. MBPs also have a similar mechanism of action to simvastatin or ezetimibe, which are statins [95]. The hydrophobic amino acids of Phe, Leu, Val, Gly and Pro in bioactive peptides may promote cholesterol homeostasis in organisms by stimulating the elimination of exogenous cholesterol during gastric emptying [96]. ABP prepared by Feng [97] reduced the solubility of cholesterol by forming micellar structures in a manner similar to the cholesterol-lowering mechanism of statins. All functional foods with MBPs may lower serum cholesterol levels. Unlike other food derivatives, they may regulate cholesterol homeostasis in organisms through different transcriptional and translational mechanisms. These mechanisms are not completely elucidated, and may be different from those of plants that have already been identified [95].

## 4. Structure—Activity Relationships of MBPs

The bioactivities of MBPs are mainly related to their molecular weight, electric charge, amino acid composition, structure and sequences of their peptides [98].

### 4.1. Molecular Weight

In most cases, the MBPs incorporated within their parent proteins are inactive, while hydrolysis releases potent peptides from the intact proteins. Most low molecular weight hydrolysates and peptides obtained by enzymatic hydrolysis exhibit more biological activities compared to their original proteins [99]. The reason is that the smaller molecular weight of peptide, the easier it is to cross the structural barrier and enter cells. Previous experiments have shown that lower molecular weight peptides bind more readily to ACE, and show positive ACE inhibitory activity when the amino acid positions at the ends of a peptide are at all sites that readily bind to the active sites of ACE-I [100]. For example, some lower molecular weight peptides of *Pleurotus abalonus* exhibited higher ACE inhibitory activity [101]. The IC_50_ value for the hydrolysate fraction (<0.65 kDa) extracted from *Lentinus edodes* is a result that exceeds those of fractions that offer greater molecular weight values according to earlier research [26]. The low molecular weight (<0.65 kDa) peptides of *Hericium erinaceus* exhibit the strongest anti-proliferative ability in the Chago-K1 cell line of lung cancer [102].

It is worth noting that with higher contents of hydrophobic amino acid residues in peptides, some larger molecular weight peptides may offer better performance. For example, the DSRA and FRAP activities, free radical scavenging capacity and lipid oxidation inhibition were found to be the highest for ABP with a molecular weight of 1–3 kDa [25,103].

### 4.2. Electric Charge

Positive charges were found to combine with the negatively charged chemokine receptors on the surface of cell membrane to depolarise the bacterial cell membrane through electrostatic interaction, rupturing the cell membrane and discharging the contents, resulting in the death of bacteria [104,105,106,107]. This mechanism by which positive charges in MBPs bind to negative charges of free radicals makes it difficult for bacteria to develop resistance, thereby significantly improving MBP antibacterial efficiency. The amino groups Lys and Arg form amino cations in aqueous solution, and exhibit excellent antibacterial properties due to their polycationic properties [66]. Antifungal peptides isolated from oyster mushrooms and fruiting bodies of *Pleurotus eryngii* were all found to be rich in lysine, which inhibited mycelial growth in *Mycosphaerella arachidicola* and *Fusarium oxysporum* [108].

### 4.3. Aromatic Amino Acids

The bioactivities of MBPs are also related to the ratio of contained aromatic amino acids. Benzene rings and indole groups of peptides containing aromatic amino acids such as Phe, Met, Tyr, Lys, Trp, Cys and His stabilised ROS by providing protons for electron-deficient free radicals, while maintaining their own stability through their internal resonance structure [109,110]. Schizophyllum commune peptides were found to be high in aromatic amino acids, and had a high capacity to scavenge free radicals [111]. Several research studies argued that the aromatic and hydrophobic amino acids are the key factors that affect ACE inhibitory activity. The results of molecular dynamics simulations suggest that QLVP [62] interacts with Lys472 of ACE through a hydrogen bond and a salt bridge, potentially exerting inhibitory effects on ACE. Compared with unhydrolysed King Boletus mushroom extract (ws-KBM), the enzymatic-bromelain King Boletus mushroom protein hydrolysate (eb-KBM) showed higher aromatic amino acids contents [91]. Eb-KBM was more active in scavenging free radicals, as assessed by hydroxyl radicals scavenging (66.0%), DPPH assays (60.9%) and the ABTS method (70.9%) [112].

### 4.4. Hydrophobic Amino Acids

MBPs are enriched in hydrophobic amino acids, which are the main components of bioactive peptides, as confirmed by mass HPLC and fingerprinting [18].

Through interactions between the hydrophobic parts of peptides and the non-polar components of cell membranes, MBPs can enter the cell to eliminate free radicals [113,114,115]. The main amino acids of CMP are hydrophobic amino acids, and hydrophilic groups of hydrophobic amino acids were found to interact with cell membranes to inhibit the growth of *Bipolaris maydis*, *Mycosphaerella arachidicola*, *Rhizoctonia solani* and *Candida albicans* [116].

Higher contents of hydrophobic and aromatic amino acids at the same molecular weight showed stronger antioxidant activities. The hydrolysate of a *Suillus pleurotus* protein exhibited antioxidative activity against DPPH^•^, ABTS^•+^ and linoleic acid oxidation at 1 mg/mL, showing high levels of surface hydrophobicity [81].

### 4.5. Amino Acid Sequence

Peptides with similar chain lengths show distinct functional activities based on the amino acid sequence at their C and N-terminal ends [117]. For example, the C-terminal tripeptide containing Trp and Tyr showed strong free radical scavenging activity, which reduced if L-His was replaced by D-His. CMP interacted most strongly with the C-terminal S2 subsite of ACE, and research will focus on its C-terminal sequence as a new antihypertensive agent [118].

Peptides MSGVAADW and SGVAADW, extracted from *Pleurotus eryngii*, secretes signal peptides. LC-HCD-MS/MS was used to analyse peptides isolated from *Lentinula edodes*, *Pleurotus ostreatus* and *Pleurotus eryngii*; they were found to contain the same “-AADW” sequence at the C-terminus [119]. All of the peptides purified from *Lentinula edodes* and *Pleurotus ostreatus* had antioxidant activity [120]. GLF and GLM isolated from *Ganoderma lucidum* were found to be more effective in their antibacterial activity against *E. coli* and *Salmonella typhi*, but the antioxidant activity of GLF was more effective [121]. GLF has hydrophobic amino acid residues (-PAPA) at its C-terminal. It can be speculated that hydrophobic amino acids are in a specific positioning at either the C- or N-terminals of constituent peptides, which may strengthen the ability to scavenge free radicals with stronger antioxidant activity.

The functional activities of MBPs are also closely related to their amino acid sequences. The effects of individual functional activities of MBPs may also differ due to differences in their amino acid sequences. In order to optimise the biological activities of MBPs, more research is needed.

## 5. MBPs in Functional Foods

Bioactive compounds such as plant extracts [122], microbial metabolites, minerals [123] and vitamins [124] may achieve desired antioxidant effects. However, most bioactive compounds are costly, and may also have side effects such as inflammation and allergies after long-term use [125]. MBPs scavenge ROS [126] from cells, and have advantages of low allergenicity [1], high bioavailability and no side effects [127,128]. In addition to being used as an alternative to natural antioxidants in food, MBPs are also used as bioactive ingredients in functional foods and nutritional products to incorporate into foods, dietary supplements and medicines [109]. MBPs have been used in a variety of functional foods including antioxidants, anticancer agents, antibacterial agents, immune boosters, cholesterol-lowering agents and hepatoprotective agents. Polysaccharide peptides (PSK) from *Lentinula edodes*; polysaccharide-binding peptides (PSP) from *Coriolus versicolor*; polysaccharide polypeptide complexes (PSPC) from *Schizophyllum commune* Fr.; and PSPC from *Tricholoma lobayense* already have a wide market as functional food ingredients [129]. Unlike animal-derived bioactive peptides, MBPs are accepted by people of all races and diets, making them an ideal alternative food for vegetarians [22]. Although research on the use of MBPs as functional food ingredients exists, their related products are not yet widely available on the market. Information on their bioactivities and interactions with other ingredients of functional foods is still lacking [130], and more detailed information is needed in the future.

## 6. Challenges and Prospects

Edible mushrooms are rich in protein. Bioactive peptides derived from edible mushrooms proteins are widely available, unique in structure, efficient and safe, and have received much interest in the field of food science and nutrition as functional food ingredients. However, only a few MBPs are known. Currently, there are still many MBPs that are unexplored compared to bioactive peptides derived from animals and plants.

Most of the existing large-scale production of bioactive peptides has the disadvantages of involving complex purification techniques. Therefore, peptide mixtures were generally chosen to study the related bioactivities. More experiments are needed in the future to verify whether the bioactive function is subject to a single action of a multifunctional peptide, or from the interaction of different peptides [99]. Thus, the challenges for deeper research in MBPs are purification and identification. In recent years, new methods such as computer simulation predictions and molecular docking have been effective tools for discovering bioactive peptides. These methods can identify target receptors and predict the affinity of peptides to receptor binding sites. These techniques effectively address time-consuming issues [104]. Semi-flexible molecular docking between shiitake mushroom peptides identified with LC–MS/MS and the crystal structure of human α-glucosidase provided peptides with α-glucosidase inhibitory activity [131]. This approach significantly improved the stability of peptides to maintain blood glucose levels, and solved the inefficient production of peptides on a laboratory scale.

The problem of poor stability has always restricted the development of peptides. MBPs are mainly exposed to the stomach, pancreas and small intestine in organisms, and are easily degraded by the gastrointestinal tract under acidic conditions [132]. Different products play different roles when the ingredients are digested. When added as ingredients, food could act as carriers for peptides. The amino acid nucleophilic side chains of some peptides react chemically with a variety of food components (e.g., carbohydrates, metals and phenolic compounds), leading to complex formations (e.g., peptide–lipid, peptide–metal, peptide–phenol) or peptide-derived products (e.g., products of the Maillard reaction) [133]. Many studies had proven that the Maillard reaction could effectively enhance the antioxidant activity of MBPs, but the effects of other compounds on biological activities need to be investigated further. Predicting the tendency of MBPs to degrade during digestion can be achieved using qualitative and quantitative analyses of specific peptides. Fingerprinting is used as one of the most effective indicators for qualitative and quantitative analyses of specific peptides to achieve quality control [134]. After analysing MBPs, it is important to avoid their interactions with other substances as much as possible. Encapsulation entails multidisciplinary technical uses such as delaying oxidation, ensuring its functionality in the subject, and avoiding unwanted interactions with other substances. Liquid self-nanoemulsifying drug delivery systems may be converted into solid free-flowing powders using probiotic- and polysaccharide-rich edible mushrooms as carriers. Studies found that edible mushrooms effectively extend the solubility of solid self-nanoemulsifying drug delivery systems [135]. Although there are few relevant studies on MBPs, more extensive research can be carried out based on edible mushrooms delivery systems.

Another challenge in the incorporation of MBPs into functional foods is that some peptides such as the γ-Glu peptide, via a calcium-sensitive receptor agonist, induce the bitterness taste mechanism, which is unpopular with consumers [136]. Current methods for reducing the bitterness of peptides in food products include extensive hydrolysis of peptide components. Peptides in the presence of bitter fragments may be filtered via membrane filtration or chromatographic separation. The bitterness of peptide components could also be masked by adding compounds that can modulate or suppress bitterness [135]. The widespread use of encapsulation could hide unpleasant odours and tastes of MBPs, improve their bioavailability, and increase the consumer acceptance of MBPs. It had been shown that liquid self-nanoemulsifying drug delivery systems can be converted into solid free-flowing powders using probiotic- and polysaccharide-rich edible mushrooms as carriers. The study found that edible mushrooms effectively extend the range of solvation in solid self-nanoemulsifying drug delivery systems [135]. Although there are few relevant studies on MBPs, extended research can be carried out on the basis of edible mushrooms delivery systems in the future. Although research has been conducted on the health benefits of MBPs, less data have been obtained from clinical trials in vivo than in vitro. Thus, in vivo and clinical trials are priorities to further investigate pathways by which peptides exhibit their biological activities, and to characterise their potential mechanisms.

In summary, future research on MBPs will focus on peptide purification and identification techniques to make them more suitable for scaled production, and to improve the bioavailabilities of MBPs without compromising the functional and organoleptic properties of these functional foods. More intensive in vivo human clinical tests of potential peptide sequences are needed in the future.

## 7. Conclusions

This paper reviewed the process of preparation and purification, the bioactivities and structure–effect relationships of MBPs, as well as the progress of research in functional foods, and discussed some challenges and prospects related to their application in functional products. This review showed that MBPs can exert antioxidant, antimicrobial and anti-inflammatory effects by modulating signalling pathways (e.g., Keap1-Nrf2-ARE, NF-κB and TNF pathways). At the same time, MBPs have significant benefits in preventing memory and cognitive deficits, hypertension, tumour, diabetes, and high cholesterol. The structure–effect relationships of MBPs were described in relation to their molecular weights, charge, amino acid compositions and amino acid sequences. MBPs can play an important role in health promotion and disease risk reduction as potential functional foods, novel food ingredients, food additives and pharmaceuticals.

## Figures and Tables

**Figure 1 foods-12-02935-f001:**
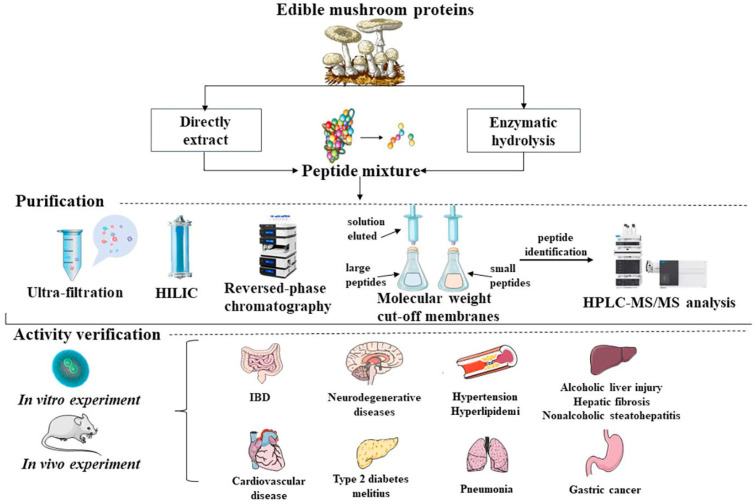
Common extraction routes, purification and preliminary bioactivities screening, in vivo and in vitro validation of biological activities of MBPs.

**Figure 2 foods-12-02935-f002:**
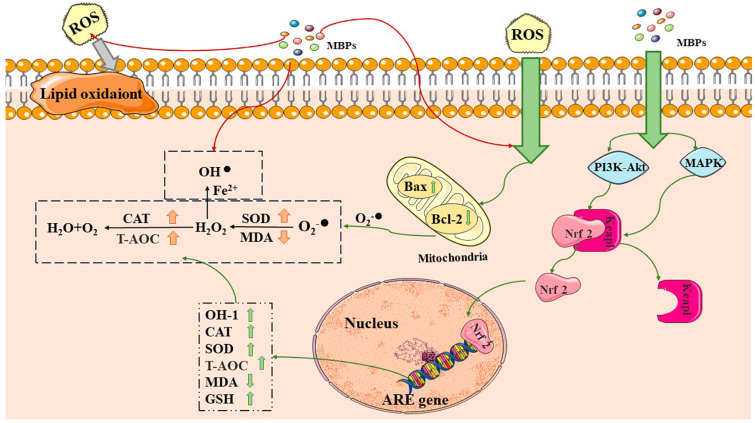
Antioxidant and anti-aging mechanisms of MBPs. MBPs may exert antioxidant and anti-aging effects by regulating Keap1-Nrf2-ARE through the PI3K/AKT and MAPK pathways. CAT: catalase, SOD: superoxide dismutase, T-AOC: total antioxidant capacity, MDA: malondialdehyde, GSH: glutathione, Bax: BCL2-Associated X, Bcl-2: B-cell lymphoma-2. Excess ROS by generating superoxide anions via Bax and Bcl-2 in cellular mitochondria. MBPs regulate ROS content by providing protons, electrons and chelating metal ions. MBPs regulate the Keap1-Nrf2-ARE signalling pathway through the PI3K/AKT and MAPK pathways. Red arrows indicate that MBPs regulate ROS content by providing protons, electrons, and chelating metal ions. The grey arrows indicate that excess ROS are harmful by generating superoxide anions. Green arrows indicate that MBPs and ROS react via different pathways within the cell.

**Figure 3 foods-12-02935-f003:**
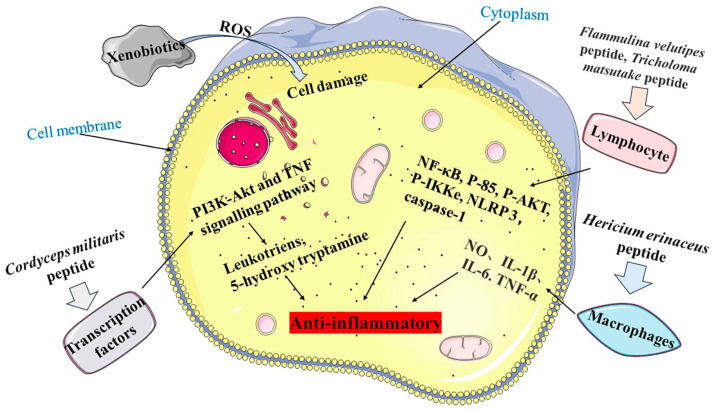
Anti-inflammatory mechanism of MBPs. Pathway 1 is *Cordyceps militaris* peptide regulating the PI3K-Akt and TNF signalling pathways to release Leukotriens and 5-hydroxy tryptamine. Pathway 2 is *Flammulina velutipes* peptide and *Tricholoma matsutake* peptide acting on lymphocytes, which in turn regulate NF-κB, P-85, P-AKT, P-IKKe, NLRP 3 and caspase-1. Pathway 3 is *Hericium erinaceus* peptide acting on macrophages to release NO, IL-1β, IL-6 and TNF-α.

**Table 3 foods-12-02935-t003:** Anti-hypertensive peptides derived from edible mushrooms.

Edible Mushroom Category	Molecular Weight and Amino Acid Sequence	Mechanisms of Operation and Value	Reference
*Hypsizygus marmoreus*	MW of 1~5 kDa Amino acid sequence: LSMGSASLSP	Hydroxyl radical scavenging activity with IC_50_ of 190 μg/mL	Kang et al. (2013) [82]
*Ganoderma Lucidum*	Peptide mixtures	Activation of angiotensin I-mediated phosphorylation of endothelial nitric oxide synthase in human umbilical vein endothelial cells, with IC_50_ of 127.9 μmol/L	Wu et al. (2019) [83]
*Tricholoma matsutake*	MW < 5 kDa Amino acid sequence: WALKGYK	With IC_50_ of 0.40 μM Inhibition of ACE activity is 63.9%	Geng et al. (2016) [52]
*Pholiota adiposa*	MW of 1~2 kDa Amino acid sequence: GEGGP	With IC_50_ of 44 μg/mL	Koo et al. (2006) [84]
*Pleurotus cornucopiae*	Peptide mixtures	With IC_50_ of 0.46 mg/mL	Jang et al. (2011) [85]
*Ganoderma lucidum*	MW of 3.35 kDa	Hydroxyl radical scavenging activity is 72.87% Superoxide anion radical scavenging activity is 72.16% DPPH radical scavenging activity is 74.21%	Girjal et al. (2012) [86]

## Data Availability

The data used to support the findings of this study can be made available by the corresponding author upon request.

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
