# Peer review of "Bioactive Peptides from Edible Mushrooms—The Preparation, Mechanisms, Structure—Activity Relationships and Prospects"

_foods, 2023, doi:10.3390/foods12152935_

Round 1

Reviewer 1 Report

Overall, this review article covers an interesting topic and makes for a good read. However, there are certain areas that can be improved to enhance its clarity and depth. Here are some suggestions:

1. Introduction:

The introduction appears to be too brief and lacks a proper introduction to the topic for readers. It would greatly benefit from incorporating more general details about bioactive peptides from other sources and highlighting the significance of MBPs. Additionally, it should explore the role of MBPs in mushrooms and explain why they hold such promising potential as resources for bioactive peptides.

2. Preparation of MBPs:

The description of the extraction method sounds like a standard procedure for protein extraction. It would be helpful to mention if there were any specific challenges or methods required for isolating MBPs. Furthermore, Figure 1 needs a more descriptive legend, and acronyms like HLIC should be defined in the text or in a list of abbreviations. Also, "Cut-off membranes" needs to be explained for better clarity.

3. Bioactivities of MBPs:

Once again, it is essential to add detailed explanations in the legend of Figure 2.

Additionally, clarify if there is any significance to the different arrow colors in the figure.

The information in section 3.1 about antioxidant activity should correspond with the graphic details in Figure 2. If some of these details are covered in section 3.4, consider combining both sections to give a comprehensive overview of the main functions, such as anti-aging activity, which is derived from the anti-oxidative and anti-apoptotic properties.

If needed, further supportive information on anti-aging activity, like DNA protection or telomerase impact, should be added. The

figure legend for Figure 3 requires more details as well. It is claimed that MBPs reduce inflammation, but the figure seems to depict an induction. Clarify if there is any specific meaning to the arrow colors that hasn't been defined yet. Additionally, use a consistent and clear form for lymphocyte and macrophage representations, either plain circles or representative graphics of their cell morphology.

By incorporating these improvements, the review article will provide readers with a more comprehensive and easily understandable exploration of the topic.

Author Response

We appreciate and appreciate your comments and suggestions, which were valuable to improve the quality of our manuscript. All of the above lines and pages appear in the revised manuscript.

Reviewer 2 Report

Title:

Bioactive peptides from edible mushrooms—the preparation, mechanisms, structure-activity relationship and applications in  functional foods

Plagiarism : Reduce the plagiarism up to 12%

Abstract: Improve the abstract, add Mushroom bioactive peptides (MBPs) in detail.

 Introduction: Add some more part related to the edible mushrooms, functional food, probiotics, natural food with latest references.

For example:

Antagonistic, anti-oxidant, anti-inflammatory and anti-diabetic probiotic potential of Lactobacillus agilis isolated from the rhizosphere of the medicinal plants

Methodology

1.     Improve the technical writing of methods by using the latest methods

Discussion:

Improve the discussion part, add latest references.

References should be journal pattern

Conclusions must be shorter and add some unique line regarding mechanisms, structure-activity relationship and applications in  functional foods

I am ready for more papers review

Author Response

We acknowledge and thank you for the comments and suggestions, which are valuable in improving our manuscript's quality. All the lines and pages indicated above appear in the revised manuscript. Please see the attachment.
